# Innovations in Electrodermal Activity Data Collection and Signal Processing: A Systematic Review

**DOI:** 10.3390/s20020479

**Published:** 2020-01-15

**Authors:** Hugo F. Posada-Quintero, Ki H. Chon

**Affiliations:** Department of Biomedical Engineering, University of Connecticut, Storrs, CT 06269, USA; hugo.posada-quintero@uconn.edu

**Keywords:** electrodermal activity, sympathetic function, EDA data collection, EDA signal processing, EDA data quality assessment

## Abstract

The electrodermal activity (EDA) signal is an electrical manifestation of the sympathetic innervation of the sweat glands. EDA has a history in psychophysiological (including emotional or cognitive stress) research since 1879, but it was not until recent years that researchers began using EDA for pathophysiological applications like the assessment of fatigue, pain, sleepiness, exercise recovery, diagnosis of epilepsy, neuropathies, depression, and so forth. The advent of new devices and applications for EDA has increased the development of novel signal processing techniques, creating a growing pool of measures derived mathematically from the EDA. For many years, simply computing the mean of EDA values over a period was used to assess arousal. Much later, researchers found that EDA contains information not only in the slow changes (tonic component) that the mean value represents, but also in the rapid or phasic changes of the signal. The techniques that have ensued have intended to provide a more sophisticated analysis of EDA, beyond the traditional tonic/phasic decomposition of the signal. With many researchers from the social sciences, engineering, medicine, and other areas recently working with EDA, it is timely to summarize and review the recent developments and provide an updated and synthesized framework for all researchers interested in incorporating EDA into their research.

## 1. Introduction

Sweat gland activity modulates the conductance of an applied current [1,2,3,4]. Such modulations produce electrodermal activity (EDA), a term that comprises the changes in electrical conductance of the skin. Increased sweating augments the electrical conductance of the skin, because although sweat contains minerals, lactic acid, and urea, it is mostly water. Thermoregulation is the primary function of most sweat glands, but those located on the plantar and palmar sides of the hand are known to be more concerned with grasping performance, rather than with temperature control. These sweat glands are more responsive to psychological stimuli rather than to thermal stimuli [4]. This phenomena is most evident in hands and feet because of the high density of eccrine glands in those areas; however, emotion-evoked sweating involves all eccrine sweat glands [5]. Therefore, EDA is believed to represent a quantitative functional measure of sudomotor activity, and consequently, an objective assessment of arousal [6,7]. Sudomotor activity is connected to the sympathetic function, and has the potential to be used for the evaluation of the autonomic function and assess the level of cognitive arousal [8,9,10]. EDA makes it theoretically possible to estimate the time and amplitude of stimuli generated from control centers in the brain by interpreting the manifestation of their arrival at the skin level, which is observable in the EDA signal [4].

The single effector model of the sweat glands is the most generally agreed-upon model for EDA. The changes in the level and phasic shifts of the EDA are the outputs of such a model. Sweat comes through varying numbers of ducts in the sweat glands at different levels, depending on the level of sympathetic arousal. The sweat ducts can be thought of as a set of variable resistors wired in parallel, which is the principle behind the single effector model. The higher the amount of sweat rises and the more ducts that are filled up, the lower the resistance in that variable set of parallel resistors. In this manner, changes in the level of sweat in the ducts produce observable variations in EDA [11].

The neurotransmitter involved in the mediation of eccrine sweat gland activity is acetylcholine, which is the primary neurotransmitter of the parasympathetic nervous system, rather than noradrenaline, which is typically associated with peripheral sympathetic activation [12]. For that reason, at one point in history, both the sympathetic and parasympathetic branches of the ANS were thought to control EDA. However, it is currently accepted that human sweat glands have predominantly cholinergic innervation from sudomotor fibers linked uniquely to the sympathetic chain [5,13]. Studies that simultaneously recorded sympathetic action potentials in peripheral nerves and EDA provide evidence for the solely sympathetic control of EDA; a high correlation between bursts of sympathetic nerve activity and the amplitude of the rapid transient events in the EDA was shown [14].

### Basics of the Signal Analysis of EDA

The most salient characteristic of an EDA signal is the occurrence of skin conductance responses (SCRs) resulting from an underlying sympathetic reaction to a stimulus. The SCRs are the rapid and smooth transient events noticeable in the EDA signal (Figure 1). At least three pathways lead to the production of SCRs: hypothalamic control, contralateral and basal ganglion influences (involves one pathway of excitatory control by the premotor cortex and another pathway of exhibitory and excitatory influences in the frontal cortex), and the reticular formation in the brainstem [13,15,16]. These pathways imply different functional roles associated with the central mechanisms: activation of the reticular formation is associated with gross movements and increased muscle tone, hypothalamic activity controls thermoregulatory sweating, amygdala activation reflects affective processes, premotor cortex activity occurs in situations requiring fine motor control, and prefrontal cortical activity is associated with orienting and attention [11,17,18]. All these processes influence the EDA signal.

Measures of the SCRs are used to evaluate a subject’s response to event-related experiments (“startle-like” stimuli) or tonic stimuli tests (like a change in condition, workload, cognitive stress, and so forth). In event-related experiments, the occurrence of an SCR is expected after the stimulus is applied. In such experiments, the SCRs are usually called the event-related SCRs (ERSCRs) [13]. Quantitative measures are obtained from SCRs by computing their amplitude, rise time (also referred to as onset-to-peak time), and other metrics. Figure 2 illustrates some of the quantitative measures available from an individual SCR. In the figure, time is relative to the stimulus and amplitude values are relative to the SCR onset level. 

The skin conductance level (SCL) and nonspecific skin conductance responses (NSSCRs) [3] are measures obtained to assess the response to a tonic stimulus. SCL, expressed in the same units as EDA (typically microsiemens (µS)), specifically refers to the overall conductance obtained from the tonic component of EDA (Figure 3), and was conceived as a measure related to the slow shifts of the EDA. A SCL is typically computed as the mean of several measurements taken during a specific non-stimulation rest period, for example, the mean of the “tonic EDA” component shown in Figure 3. The non-specific NSSCRs are the number of SCRs in a period of time and are considered a tonic measure because they cannot be linked to a specific stimuli, but are the result of spontaneous fluctuations in EDA in the presence of an ongoing sustained stimulus over a period of time. 

Despite the source that caused the SCRs (specific to a stimulus or spontaneous), they are characterized by a rise from the initial level to a peak, followed by a decline [19]. When caused by a stimulus, the onset of the SCR is typically between 1 and 5 s after the delivery of the stimulus [3]. The amplitude of the SCRs (conductance at the peak relative to the conductance at the onset) can reach several µS. A minimum of 0.05 or 0.04 µS is typically set as a threshold to define a significant SCR to avoid incorrect measurements caused by movement artifacts, the noise level of the equipment, and experimental conditions [3]. The time from the onset of the SCR to the peak, termed the rise time (Figure 2), normally varies between 0.5 and 5 s [20]. The spectral content of EDA is mostly confined to 0.045–0.15 Hz [21]. Exercise increases the spectral content of EDA, exhibiting spectral content at about 0.37 Hz when subjects perform vigorous-intensity exercise [22].

## 2. Methods

We reviewed advances in EDA data collection and signal processing techniques from the last 10 years. The first author searched PubMed, Web of Science (WoS), and the Scopus database to identify research articles and conference papers published in the last 10 years. As EDA is also referred to in some studies as “galvanic skin response” or “skin conductance response,” these keywords were also used in the search (TITLE (“electrodermal activity” OR “galvanic skin response” OR “skin conductance”) AND DOCUMENT TYPE (article) AND PUBLICATION YEAR > 2008). From the results of the search in the three databases, we extracted studies in three categories of our main interest: (1) EDA recording devices and electrodes, which included all the studies that investigated instrumentation, technologies, and sensors for EDA collection development and testing; (2) processing techniques for EDA signals, which included the signal processing techniques developed to create novel, sensitive, and quantitative measures based on the EDA; and (3) EDA quality, which included studies that evaluated the reproducibility and consistency of measures of EDA, as well as techniques for managing motion artifacts and noise.

## 3. Results

The search found 365 entries in Pubmed, 377 in WoS, and 417 in Scopus. From these search results, 16 manuscripts were classified as describing EDA recording devices and electrodes, 28 reported on processing techniques for the EDA, and 9 studies investigated EDA quality. For context, the previous work has been included, as described in [13].

### 3.1. EDA Data Collection: Recording Devices and Electrodes

#### 3.1.1. Endosomatic versus Exosomatic Recordings

It is possible to collect EDA signals without an external source of electricity, in which case it is called endosomatic recording. In this case, the voltage between an active site and a reference electrode at a relatively inactive site is collected. Endosomatic devices are thought to be simpler as they require only a high input impedance amplifier (>10 MΩ); however, they require an amplifier gain and a floating reference to measure the potential difference between the two electrodes. The signal obtained with this method is termed the skin potential response. It has a direct relationship to SCRs [11]. However, the skin potential response can be monophasic positive, monophasic negative, biphasic, or triphasic. This complexity hinders the scoring and interpretation of the signal [3]. This has limited the use of endosomatic recordings in recent studies.

Most EDA devices use an exosomatic approach, in which an external constant current or voltage source is applied via electrodes on the skin [3]. Exosomatic devices measure the modulated current or voltage, depending on whether the constant source is a voltage (most typical) or a current, to compute the skin conductance using Ohm’s law. To prevent endosomatic contamination of exosomatic measurements, the latter devices typically use a reference common to output and input that makes the voltage difference independent of a reference electrode position. The constant source for exosomatic recordings can be either a direct current (DC) or an alternating current (AC) source.

#### 3.1.2. DC versus AC Sources in Exosomatic Recordings

Although either a constant current or constant voltage can be used for exosomatic recordings, the latter is widely recommended [23]. The collection of EDA data using a DC voltage source is the most common method in psychophysiological applications, given the simplicity of its implementation. This approach can provide both tonic and phasic components of EDA, and can be technically achieved by means of operational amplifiers [24]. The main disadvantage of DC-source (voltage or current) devices is the polarization of the electrodes and the counter emf generation at the electrodes, which corrupts the collected EDA, even when using nonpolarizing Ag/AgCl electrodes. 

Exosomatic recordings using an AC voltage or current source require more sophisticated instrumentation than for the DC approach. Given that conductance and capacitance (resulting from the capacitive properties of the stratum corneum) are physically in parallel, using constant amplitude voltage excitation is recommended (the resulting current is measured). To separate the components representing the conductive and capacitive contributions, a phase-sensitive rectifier is typically used. The foremost advantage of AC-source devices compared to DC-source devices is the avoidance of electrode-polarization issues or a counter emf [3]. Although AC-source devices are highly recommended over DC-source devices, both types of EDA devices are widely used. Undertaking more research to support the superiority of the AC method to DC methods has been suggested in previous studies [3].

#### 3.1.3. Basic Considerations on Electrodes for EDA Measurements

Due to the nature of EDA and the location where the signal is acquired, recording EDA signals may require special electrodes, electrode gels, and recording devices that differ from the ones used for other psychophysiological measures. Electrodes are an important factor in the quality of the EDA measurements. The use of the same material for the two EDA electrodes is very important for EDA data recording, mainly in DC-source devices, because a difference in metals will introduce a potential difference. The use of silver-silver chloride (Ag/AgCl) electrodes is the standard for EDA recording. Ag/AgCl electrodes help minimize both the polarization of the electrode and the bias potential between the electrodes. The hydrogel improves the signal quality by lowering the impedance that exists at the electrode–skin interface. However, it reduces the shelf life of hydrogel electrodes as the hydrogel layer that exists in the skin–electrode interface degrades with time due to dehydration. The high impedance produced by dehydration leads to an impairment in the quality of the signal and to an increased signal corruption caused by motion artifacts and noise.

Dry electrodes can potentially avoid the signal degradation issues and inventory shelf-life management. Metallic electrodes are widely used, but carbon electrodes have gained some popularity for EDA recordings [3]. In general, the main advantage of dry electrodes over hydrogel electrodes is that they do not degrade over time (lacking hydrogel), and thus can potentially collect better long-term recordings. As dry electrodes do not have a shelf-life limitation, they also save the expense of managing an inventory and scrapping obsolete inventory. 

A big concern in EDA measurement is the polarization of the electrodes. In exosomatic DC-source devices, the electrodes are polarized by electrolysis because they carry a DC current and become the anode and cathode in an electrical system. Reusable EDA electrodes, which are often commercially available, are much more expensive than disposable AgCl electrodes with very thin layers of silver chloride. In reusable electrodes, it has been observed that even with a low DC current, the charge can become sufficiently high to remove part of the AgCl layer at the cathode after prolonged use, and increase the layer at the anode, producing large bias voltage levels [23]. 

#### 3.1.4. Advances in Technologies for EDA Data Collection

Multiple studies have reported advances in EDA data collection in recent years, mainly supporting the feasibility and superiority of AC devices over DC devices. A study reported the development of a device to enable the DC potential and AC conductance to be measured simultaneously at the same skin site by using a small AC current in a monopolar system [25]. They found the skin potential responses to be more robust to movement artifacts. The developed instrument can also detect whether the indifferent electrode is connected on an inactive skin site. Another study looked at the feasibility of high-frequency alternating current for collecting EDA measurements [26]. They collected skin admittance measurements from 1 Hz to 70 kHz and used the interval 500 Hz to 10 kHz to fit a Cole model to the measured skin admittance. The method overestimated the skin conductance by about 20%, suggesting that the proposed skin admittance method is not suitable for the estimation of a low-frequency skin conductance level, but the method can still be used to collect the variations of EDA at higher sampling rates. A more recent study aimed toward directly comparing AC and DC measurements of EDA, following the suggestion from Boucsein et al. [3], in order to validate the AC method by comparing it to a standard DC method [27]. They found a voltage of 0.2 V to be sufficient for DC recordings. Their main observation was the excellent agreement they found between a 20 Hz AC method and a standard DC method. These studies provide further support for the validity and superiority of the AC recording methodology. 

Electrodes for specific applications of EDA measurement have also been developed recently. Stainless steel electrodes are mechanically strong and avoid the risk of corrosion, which makes them suitable for many applications including wearable devices. However, this material is more prone to polarizing than AgCl electrodes. Alternatively, carbon can be used instead of metal for both the wires and the electrodes because this material not only reduces polarization, but can also make the electrodes radiotranslucent, and magnetic resonance imaging (MRI)-compatible [3]. Some groups have developed instrumentation capable of collecting EDA measurements that is compatible with magnetic technologies, aiming to enable the simultaneous measurement of central and peripheral nervous activity [28,29]. For example, a device meant to avoid the mutual interference effect between magnetic resonance and EDA devices was developed in Lim et al. [28]. The system was designed using only analog elements in order to remove any possible effects that digital devices cause on magnetic resonance images. They used carbon electrodes instead of Ag/AgCl electrodes to minimize the induced noise. Another group developed a low-cost system for recording EDA from humans in a magnetically shielded room simultaneously with magnetoencephalography [29]. The group used Ag/AgCl electrodes in a “sensors’ plate” used to immobilize participants’ distal sites of the fingers. These devices are important for neuroscience applications, allowing for the simultaneous assessment of autonomic and central nervous system activity.

Some groups have developed electrodes that can potentially perform in wearable devices. A group proposed a highly wearable EDA sensor based on flexible conductive foam as the sensing material and designed it to be easily attached and detached from clothing [30]. The sensor was tested on the back of the user and provided reliable measurements. Another group developed dry carbon/salt adhesive electrodes that were found to have no significant differences in amplitude, onset-to-peak time, and onset time when compared to Ag/AgCl electrodes, showing the feasibility of the mixture for collecting EDA signals [31]. Furthermore, carbon-based electrodes can potentially be cheaper to fabricate compared to Ag/AgCl electrodes [31,32]. More recently, the design and feasibility of breathable and flexible dry Ag/AgCl electronic textile electrodes for EDA data collection were studied [33,34]. Interestingly, they also reported that a minimum of about 140 sweat glands needed to be covered by the electrodes in order to maintain functionality. These flexible conductive foam, carbon/salt adhesive, and Ag/AgCl electronic textile dry electrodes are potentially suitable for wearable applications. More research is necessary to evaluate the quality of the signal collected with these sensors in long-term applications. 

Other research groups have developed novel devices aiming to foster wearable technology to collect EDA data. A group developed a glove with embedded circuitry and based on conductive fabric sensors that is capable of collecting EDA and pulse waves [35]. Later, an unobtrusive wrist-worn integrated sensor was tested in a study for ambulatory, long-term, continuous assessment of EDA [36]. They also found evidence that the distal forearm is a viable sensor location for EDA measurements. This location could provide a more comfortable, continuous assessment of EDA. Attempts have been made to develop miniaturized instrumentation for collecting EDA data. A group of researchers developed a miniature EDA monitor (7.2 cm × 3.8 cm × 1.2 cm) and algorithm for investigating hot flash events [37]. The device was reported to be capable of collecting data for seven days without external wires. Another study evaluated the feasibility of an EDA ring prototype by comparing the similarity of signals between a prototype of the wearable Moodmetric EDA Ring and a laboratory-grade skin conductance sensor in a psychophysiological experiment [38]. The prototype ring seemed to be a promising wearable tool for future studies. Recently, the design and implementation of an ultra-low resource EDA sensor for wearable applications incorporated a compression method [39]. Although theirs is a DC-source topology, the system and compression method could improve the functionality of low-resource microcontrollers. Lastly, it is worth mentioning a proposed method to measure EDA contactless (over the clothes) using a 5 kHz current source [40]. Their results support the feasibility of such a technology.

### 3.2. EDA Signal Processing: Techniques for Performing Data Decomposition and Analysis

#### 3.2.1. Tools for Scoring EDA and Recording Contextual Information

The scoring of SCRs was done manually for many years. The main benefit of manual scoring is the ability of the experimenter to get a close trial-by-trial inspection of waveforms to make sure that individual SCRs are physiologically related to an event of interest. However, manual scoring is a cumbersome and subjective process. Aiming to facilitate the analysis of emotion-related data, several computer algorithms have been implemented. A group of researchers developed a tool that offers a user-friendly interface for pre-processing and assistance with the peak scoring (latency, rise time, amplitude, and duration) of individual SCRs [41]. More recently, an application named “Autonomate” was developed to automate the manual scoring of ERSCRs. The software offers tools to account for overlapping SCRs and other common problems that introduce bias in manual scoring, such as consistency in applying response criteria [42]. The spontaneous fluctuations, those that are not specific to any identifiable external event, complicate the use of automatic approaches and some researchers still prefer manual scoring tools for their event-related studies.

The availability of wearable sensors for EDA data collection has increased opportunities for the technology to be used in studies looking at the effects of experiences and environment on physiology. In such scenarios, contextual information is required. An architecture and implementation of a system for acquiring, processing, and visualizing EDA and other biophysiological signals along with contextual information was developed [43]. Their results indicate that the system allowed the users to properly annotate contextual information to be used in the analysis of biophysiological signals.

#### 3.2.2. Automatic Scoring of EDA

Methods for automatically measuring the SCRs using mathematical models are attractive from a theoretical and procedural standpoint. There are many studies that have examined automatic ways to count spontaneous SCRs, extract amplitude or other measures of a single causal SCR, and manage the superposition of SCRs. The main challenge is the frequent occurrence of a second SCR before the completion of a given SCR. 

##### Tonic/Phasic Decomposition of EDA

Aiming to resolve the problem of overlapping SCRs, some researchers proposed a method inspired by the analysis of blood-oxygen-level-dependent responses in functional magnetic resonance imaging to detect and analyze SCRs [44]. The method uses a linear convolution model and makes use of the full SCR instead of a traditional peak-scoring approach. The code was made available as SCRalyze, but newer versions are incorporated into another tool [45]. Previous versions are still available [46]. Another group developed a tool to analyze individual SCRs by assuming that each SCR can be regarded as the output of a linear invariant filter [47]. A large part of the variance in SCRs can be explained by individual-dependent response functions. The group collected EDA data at different sites on the body so that the results they obtained were not confined to palmar recordings. All model-based analyses of EDA assume that SCRs are generated by a linear and time-invariant system. These assumptions were systematically tested in a subsequent study [47]. Based on the results, both assumptions seem to hold. They also used the convolution model approach to quantify sympathetic arousal rather than recovering the sudomotor nerve activity, as attempted by other studies [48]. The amplitude of event-related SCRs is used to infer sympathetic arousal as SCRs are generated by sweat secretion initiated by distinct bursts of sudomotor nerve activity. The model uses the time-integral of the measured conductance as a measure of sudomotor bursts’ amplitude and frequency. This measure proved to be a sensitive predictor of autonomic arousal. Later, they proposed a dynamic causal model to describe the EDA signal given an underlying sudomotor nerve activity. The inversion of the model can describe the sudomotor nerve activity for the defined model and the observed EDA. This method does not rely on any given stimulus time as the time, duration, and amplitude of the sudomotor activity can be estimated directly from the EDA [49,50].

Another group proposed a non-negative deconvolution method to separate EDA into continuous tonic and phasic components, and obtain discrete compact responses [7,51]. The resulting decomposition of single non-overlapped SCRs allows for measuring the response parameters more precisely. This tool is available in Ledalab [52]. SCRalyze and Ledalab model-based methods were compared in terms of their sensitivity in recovering sympathetic arousal from SCRs analysis [53]. SCRalyze exhibited a better performance than Ledalab at distinguishing between pairs of sympathetic arousals that are known to be different. 

An improvement to the model-based analyses was proposed in Bach et al. [54]. The algorithm makes use of the between-subject SCR shape variability and high pass filtering (0.05 Hz cut-off frequency) of EDA. They also found that non-linear models better reconstructed the signals but had a lower predictive validity compared to a constrained individually-optimized response function. Another study proposed a knowledge-driven method to represent EDA [55]. The EDA-specific dictionaries were used to model both the tonic component and the SCRs contained in the phasic component. A greedy sparse representation technique was used to decompose the signal into a small number of atoms from the dictionary. The method performed well for signal reconstruction, compression, and information retrieval. A matching pursuit algorithm was presented as a fast inversion method for inferring sympathetic arousal from the fluctuations in the EDA signal [56]. This algorithm was able to approximate the true sympathetic arousal, up to 10 spontaneous fluctuations per minute, in simulated data. The computation of this approach is about three orders of magnitude faster than the dynamic causal model. 

Another group proposed a model that describes the EDA as a linear combination of the tonic component, the phasic component, and noise (which incorporates the error of the model, artifacts, and other measurement errors). The phasic component results from the convolution between an infinite impulse response function and a sparse, non-negative sudomotor nerve activity driver. Based on these assumptions, they extracted the tonic and phasic components of the EDA using a convex optimization problem, which was constrained by non-negativity and sparsity of the sudomotor nerve activity [57]. The algorithm, called cvxEDA, showed good results in decomposing the EDA signal into tonic and phasic components in simulated and experimental data with different levels of noise. The algorithm requires the user to adjust two parameters, α and γ, to penalize the phasic and tonic components of the decomposition. A larger α reaches a sparser phasic component, while a higher γ produces a smoother tonic component. In this approach, the known inter- and intra-subject variability could be addressed by using different impulse response functions for a specific subject and/or condition. The algorithm, available at [58], has been heavily used in many applications. The low computational cost may enable the use of cvxEDA in wearable devices.

The explicit incorporation of motion artifacts into the tonic component, called the baseline signal, to produce a more realistic EDA model was proposed later [59] to overcome the difficulty where changes in the position of wearable sensors due to movement may lead to rapid changes in signals. The decomposition of EDA into the baseline, the phasic component, and noise was achieved by using a sparse deconvolution and a proposed compressed sensing-based decomposition. They modified the compressed sensing tools to extract the SCRs using a concise optimization program and corresponding recovery error bounds. The results obtained after testing on simulated and experimental data suggest that this approach has a higher accuracy for SCR detection than the previously published tools. 

Similarly, another group proposed using the Hartley modulating function to assure convexity of the optimization formulation for estimating the number, timing, and amplitudes of underlying neural firing from EDA signals, and using Kaiser windows with different shape parameters to emphasize the significant spectral components [60]. This approach was meant to maintain a balance between noise filtering and enhancing the relevant information in the EDA data. This algorithm outperformed cvxEDA and Ledalab in identifying the neural stimuli. Another fully automated approach for tonic/phasic decomposition of EDA data based on non-negative sparse deconvolution and multiscale modeling of SCRs was proposed [61]. The algorithm, called SparsEDA, is reported to be faster and more efficient (it works for any sampling rate and signal length) and more interpretable (the phasic component obtained is highly sparse) than cvxEDA or Ledalab. SparsEDA, which is publicly available [62], was tested on data from 100 subjects to confirm its advantages and performance. The main value of SparsEDA is that it allows for the fully automated extraction of SCRs from large and small EDA segments, which is key for wearable applications. 

##### Spectral Analysis of EDA

A very different approach for EDA data processing based on spectral analysis was proposed recently [21]. This approach was motivated by the spectral analysis of heart rate variability (HRV), which is used to assess the dynamics of the autonomic nervous system by computing the power spectra in two main bands. It is known that the high-frequency (0.15 to 0.4 Hz) components of HRV are solely influenced by the parasympathetic system and the low-frequency components (0.045 to 0.15 Hz) are influenced by both the sympathetic and parasympathetic nervous systems. In the presence of several stressors, a significant increase in the spectral power of the EDA was found in the same band as the low frequencies of HRV, which are known to be, at least in part, controlled by the sympathetic nervous system. The expanded frequency range of 0.15–0.25 Hz (accounting for an additional 5–10% of the spectral power of EDA) was proposed as an index of sympathetic control based on the power spectral analysis of EDA, termed EDASymp. This index was sensitive to stress in a similar fashion to time-domain measures (i.e., SCL and NSSCRs) in response to most stimuli but was even more sensitive to the stress induced by the cold pressor test. The sensitivity and consistency of the spectral analysis index of the sympathetic control were subsequently improved by using a time-varying spectral analysis approach [63]. The new index of sympathetic control (termed TVSymp), incorporating the components between 0.08 and 0.24 Hz, was found to be highly sensitive to orthostatic, cognitive, and physical stress, exhibiting a higher between-subject consistency than did other measures of EDA, including SCL, NSSCRs, and EDASymp. The change in the high boundary of the spectral band containing the power of EDA under physical activity was explored in a further study [22]. The evidence suggested that the boundary is about 0.37 Hz under vigorous-intensity exercise. EDA, as a marker of sympathetic control, has also been proposed for improving the assessment of the sympathovagal balance [64]. Time-varying power in the EDASymp band was used to assess sympathetic activity, and instantaneous parasympathetic dynamics were measured using a point-process model for heartbeat dynamics. Results of the cold pressor testing showed that the increase in the proposed sympathovagal marker is more consistent and showed a higher statistical discriminant power compared to the standard low-frequency/high-frequency (LF/HF) ratio.

Another study introduced the EDA-gram, a multidimensional fingerprint of the EDA signal [65]. The EDA-gram was inspired by the spectrogram and is based on the sparse decomposition of EDA using a set of dictionary atoms. The tonic and phasic atom selection is knowledge-driven. The spectral dimension of the EDA-gram depicts the width of the selected dictionary atoms, and the intensity is the atom coefficients, which represent in turn the amplitude of the SCRs. The results of testing this approach suggest that some features derived from the EDA-gram can differentiate between arousal levels and stress type because it accentuates the signal fluctuations. 

##### Other Approaches for Decomposition and Scoring of EDA

A point process approach for characterization of the EDA signal was introduced recently [66]. This technique aims to overcome the lack of statistical models for analyzing the occurrence of SCRs in EDA based on physiology that consider the stochastic structure of the signal to provide insight into the underlying autonomic dynamics. The point process framework was used in conjunction with an inverse Gaussian distribution to track the instantaneous dynamics of EDA. Although the reported results are based on data from only two healthy volunteers under controlled sedation with propofol, they provide preliminary evidence that point process models that consider the physiology of the phenomena and are constructed upon the specific statistical structure of the SCRs have potential to track instantaneous activations of the sympathetic nervous system. 

The state-space approach has also been used for EDA data analysis. Similar to other studies using state-space models to estimate an unobserved neural state from physiological data, in a particular study, the authors related stress to the probability of occurrence of a specific SCR (a phasic driver impulse) in the EDA [67]. The study showed promising results for extracting measures to continuously track a stress level elicited by cognitive and emotional stress, as well as relaxation, using EDA data collected by wearable devices.

A new tool to perform SCR detection from EDA data that accounts for respiration was made available recently in a MATLAB toolbox called Breathe Easy EDA (BEEDA) [68]. Irregular respiration and deep breaths usually cause fluctuations in the EDA signals that can be confounded with SCRs, making their accurate detection more difficult. BEEDA facilitates visual inspection of EDA signals, allowing for the deletion of respiration artifacts, and trough-to-peak measurements of individual SCRs. It also includes functionality for EDA data decomposition into tonic and phasic EDA components, and artifact identification. The tonic component is assessed using the mean and standard deviation of the segments of EDA.

### 3.3. EDA Quality

A key concern about the SCL and NSSCRs is that these indices are highly variable between subjects [69]. In addition, periodic shifts in the background SCL (e.g., a DC shift) could be important if they appear to occur in conjunction with specific components of the experiment, and only a visual analysis would reveal the difference between an SCR and unimportant drift factors (artifacts) [70]. Traditionally, obtaining NSSCRs required an observer to count SCRs, which was difficult if the EDA measurements were affected by motion artifacts. In particular, artifacts like patient motion, temperature fluctuations, and noise can be confounded with SCRs. The inability to distinguish these artifacts from real SCRs limits the reach of ambulatory EDA data analysis and interpretation. The manual identification and removal of artifacts from EDA data is possible, but it is a time-consuming task. The traditional approaches for dealing automatically with noise and motion artifacts in EDA include low-pass filtering (typical cutoff frequency = 1 Hz), exponential smoothing, and removing corrupted signal segments (e.g., connect endpoints using a spline) [71,72].

#### 3.3.1. Motion Artifacts Detection and Correction

Several studies have focused on motion artifact detection and/or correction of the EDA signal. A machine learning algorithm for the automatic detection of artifacts in EDA signals was presented in Taylor et al. [73]. Their approach was tuned to detect motion artifacts in 5-s segments using features extracted from the amplitude, the first and second derivatives of the EDA, and wavelet coefficients. They tested several machine learning strategies for classification, including neural networks, support vector machines, naïve Bayes, and others. Although their overall accuracy was not very good, the results are promising because they proved the feasibility of detecting motion artifacts in EDA data. The resulting tool is available online [74]. Another approach for artifact and noise suppression was presented using a biophysical model for EDA implemented with an extended Kalman filter [75]. The filter was tested on real and simulated noise and artifacts. Their results suggest that noise and artifacts can be suppressed while obtaining an estimate of the sudomotor nerve activation by using this approach.

A method for removing motion artifacts from EDA using a stationary wavelet transform was proposed [76]. The wavelet coefficients are modeled as a Gaussian mixture distribution corresponding to the underlying tonic and phasic components of EDA. The denoising procedure uses a stationary wavelet transform to expand the EDA signal (presumably contaminated) into multiple levels of scaling and wavelet coefficients. Then, the threshold limit within each time window at each level is adaptively selected based on the statistical estimation of the wavelet coefficients’ distribution. Such a threshold is used on the wavelet coefficients of all levels. Finally, the inverse wavelet transform is applied to the thresholded wavelet coefficients to obtain the denoised EDA signal. Compared to traditional approaches, this method exhibited a higher performance in reduction of artifacts. 

Curve fitting and sparse recovery methods have been used to automatically identify and remove artifacts in EDA data [77]. Specifically, these researchers used an orthogonal matching pursuit sparse recovery algorithm to address the artifacts and perform SCR detection. They tested the effect of different filters (0.35, 0.5, and 1 Hz) on the determination of SCR morphology and artifact detection, and reported a maximum accuracy of around 80% for correctly labeling corrupted EDA data as artifacts.

Another simple, transparent, and flexible method for the automatic quality assessment of EDA data was reported [78]. The algorithm uses four rules to identify corrupted EDA data: (1) data out of range; (2) too-rapid changes; (3) subject not wearing the device, as suggested by temperature sensor; and (4) data surrounding the segments (i.e., transitional) were identified as invalid via the preceding rules. The method exhibited a high sensitivity (91%), specificity (99%), and overall accuracy (92%) when compared to labels provided by experts who inspected the data visually. The method was intended for ambulatory data but it can be used to enhance the quality and reproducibility of EDA analyses in general. The software is freely available online [79].

#### 3.3.2. Variability and Repeatability of Measures of EDA

The deployment of indices of EDA in clinical settings and daily use requires repeatable and robust measures. High inter-subject variability of time-domain measures of EDA has been reported [21,69]. Specifically, the SCL and NSSCRs exhibited higher coefficients of variation (i.e., the standard deviation of measurements divided by the mean) and lower degrees of consistency (assessed using intra-class correlation coefficient) between subjects undergoing cognitive, postural, and physical stress when compared to spectral indices. Recent studies have looked into the intra- and inter-subject variability of EDA measures [80]. A recent paper explored the latency of SCRs (stimulus to SCR onset time), as this characteristic is thought to be a major indicator for defining an appropriate response. The latency of SCRs was investigated for tactile, auditory, and visual stimuli, as well as its fluctuations over the course of a learning experiment. The results suggested a modality-specific latency of the SCRs. They found evidence for gender and cognitive effects while exploring individual differences in SCR latencies. This suggests that the inter- and intra-subject variability of SCR latencies may contain information, besides serving as criteria for defining response windows. In another recent study, the five-day reproducibility of measures of sympathetic control based on heart rate variability and EDA was assessed [81]. They tested the consistency and reproducibility under orthostatic and cognitive stress in highly controlled conditions without environmental causes of variability. Indices obtained from heart rate variability and the time-varying spectral index of EDA (TVSymp) exhibited higher consistency during the orthostatic test compared to other EDA measures. Indices of EDA exhibited a higher consistency overall in response to cognitive stress when compared to HRV. TVSymp was the most reproducible measurement on average (lowest coefficient of variation and highest intra-class correlation coefficient) for both types of stimuli.

## 4. Discussion

EDA has a long history in psychophysiological research since the studies of Vigouroux in France in 1879. During the first years of use of the technique, it was used to evaluate EDA as a response to mental (e.g., emotional, cognitive) stress. More recent observations showing that EDA varies with the state of sweat glands in the skin, which are controlled by sympathetic nervous activity, initiated the practice of using the technique as an indication of not only psychological but also general sympathetic arousal. Despite the wide acceptation of this concept, using EDA for pathophysiological assessment is rather novel. 

Literature suggests the practical superiority of the constant-amplitude AC-voltage exosomatic method for reliable EDA data collection. This approach avoids the complexity of collecting signals using endosomatic approaches and the error in the EDA signal introduced by electrode polarization. Nevertheless, the constant-amplitude AC-voltage exosomatic method allows for the collection of skin potential, skin conductance, and skin susceptance (the imaginary part of admittance) at the same skin site, which is of interest for some groups attempting to assess the functioning of the sympathetic nervous system [82,83]. The constant-amplitude AC-voltage exosomatic configuration is advisable for the development of future applications based on EDA. 

The development of wearable technologies capable of reliable EDA data collection and analysis is a relevant research topic. It involves the type of sensor, the measuring site, the signal conditioning, motion artifact detection and correction, and the suitability of the resulting measures of EDA. Furthermore, different scenarios, such as sleeping, exercising, driving, and others, have specific requirements, such as quality and data length, to provide the intended measures based on EDA. The feasibility of wearable devices to provide the required chain of capabilities (collection–processing–diagnosing) should be tested in such scenarios.

Tonic/phasic decomposition is the most-pursued task for EDA signal processing. The scientific relevance of tools for this task relies on their ability to detect the underlying sympathetic driver that produces a specific phasic shift. If such a task is accurately achieved using EDA, one could use the model to objectively measure subjects’ reactions to specific stimuli or significant situations (e.g., public speech, advertisements, pain, and so forth). Several implementations of the tools are available. However, the main practical constraints of such methods are the feasibility of implementing the mathematical computations in wearable devices and the requirement of subject-specific tuning. In particular, the convex optimization approach (cvxEDA) and the sparse deconvolution approach (sparsEDA) are relatively fast techniques that can be implemented in a wearable device given their low computational cost. Nevertheless, both tools require setting a group of parameters (four or more). If the default values for the parameters are used, the results of the decomposition vary highly depending on the subject and the application. There is no congruency on the results between the decomposition methods. For illustration purposes, in Figure 4 we have included the tonic/phasic decomposition of a given EDA signal using the cvxEDA [57], the sparsEDA [61], the continuous decomposition analysis available in Ledalab (CDA-Ledalab) [84], the discrete decomposition analysis also available in Ledalab (DDA-Ledalab) [84], and the dynamic causal modeling available in psPM (DCM-psPM) [49]. Besides the difference in computational time, each algorithm provides a very different estimation of the tonic and phasic component. Machine learning and deep learning could help with this task. If a well-established definition of what the tonic and phasic components should be comprised of existed, a model could be trained to perform such a decomposition. 

A potential application of EDA is to use it jointly with heart rate variability to develop indices of autonomic function. For instance, EDA can be used to adjust the bands for spectral analysis of heart rate variability under exercise. Also, adaptive filtering approaches are able to use the information in EDA to obtain more accurate, sensitive, and specific indices of autonomic control and balance. The heart rate variability is known to have a nonlinear relationship with the autonomic control [85]. For its part, the order (i.e., linear or nonlinear) of the interplay between EDA and the sympathetic tone must be determined before we can obtain suitable indices of autonomic control combining EDA and heart rate variability.

Furthermore, ways to increase the specificity of EDA for the assessment of the sympathetic nervous system in clinical applications need to be explored. The correct diagnosis of many diseases requires more sensitive and reliable measures of sympathetic function. An important example is diabetic cardiovascular autonomic neuropathy, which is present in at least 25% of diabetics. It is a remarkable example of the need for sensitive measures of sympathetic tone [85,86] because the gold standard procedure for sympathetic assessment for this disease is the cardiovascular autonomic reflex test [87], which has a low sensitivity (50%) [88]. New quantitative and accessible methods for assessment of the sympathetic nervous system are needed. 

Future applications of EDA may include the multi-parameter approach (e.g., precision health) based on wearable sensors for assessing diseases that affect the autonomic control (stress, neuropathies, pain, and so forth). We envision those tools to incorporate artificial intelligence for obtaining and selecting features, as well as for estimating the level of progress of disease. A different application would be the detection of sleep drowsiness, which could be used, for example, to alert drivers when they are too tired to be driving safely because their bodies are showing low levels of responsiveness. The simplicity of the circuitry to collect the signal and the absence of parasympathetic interference make the EDA a valuable source of information and a desirable target for many applications. Several wearable devices already incorporate EDA sensors [89] but their use is limited to functions like detecting whether the user is wearing the device, or basic analysis, such as merely computing the conductance level.

## 5. Conclusions

The best electrode type for EDA data collection highly depends on the application. Although Ag/AgCl hydrogel electrodes provide the most reliable signals, they are impractical for wearable applications that require long-duration reliability. More research is required to validate the feasibility of stainless steel and carbon electrodes to provide comparable signal quality. Developing reusable electrodes that are reliable for long-term applications can boost the deployment of EDA applications. As for the electronic devices, AC-voltage devices seem to be the most appropriate approach overall. Despite their higher circuit complexity, they avoid the issues caused by electrode polarization. Current techniques for automatic analysis of EDA signals still require improving their subject independency and their ability to function without setting any external parameters to make the technique feasible for ambulatory settings. Furthermore, alternative techniques like spectral analysis have emerged as potential tools for the analysis of EDA. Some tools for detecting and removing corrupted segments from EDA signals are available but their sensitivity needs to be further evaluated. The consistency of the measures derived from EDA is still a matter of concern, specifically in the presence of motion artifacts.

## Figures and Tables

**Figure 1 sensors-20-00479-f001:**
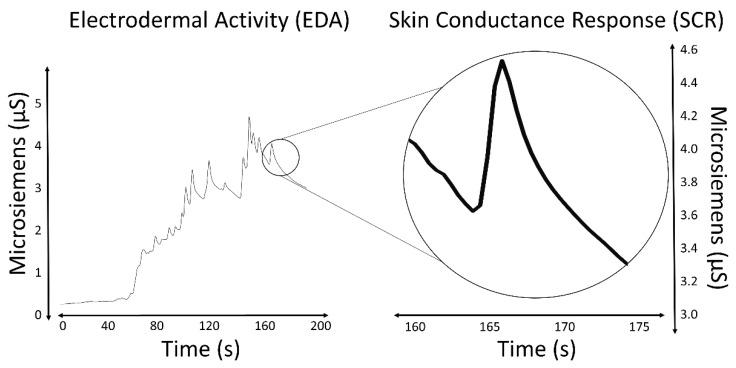
EDA signal and an isolated SCR.

**Figure 2 sensors-20-00479-f002:**
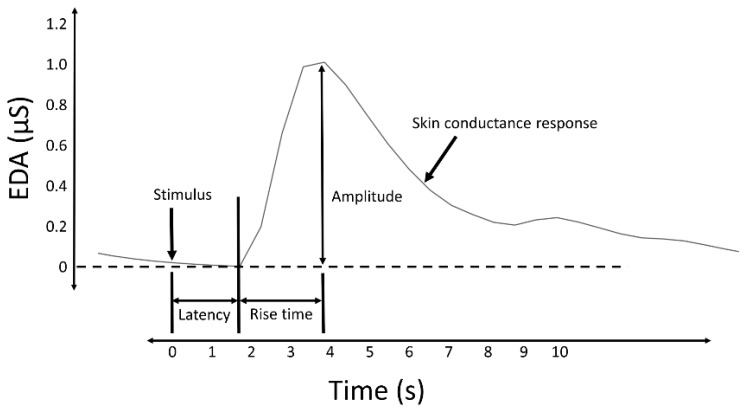
A typical skin conductance response (SCR) and illustration of some derived measures.

**Figure 3 sensors-20-00479-f003:**
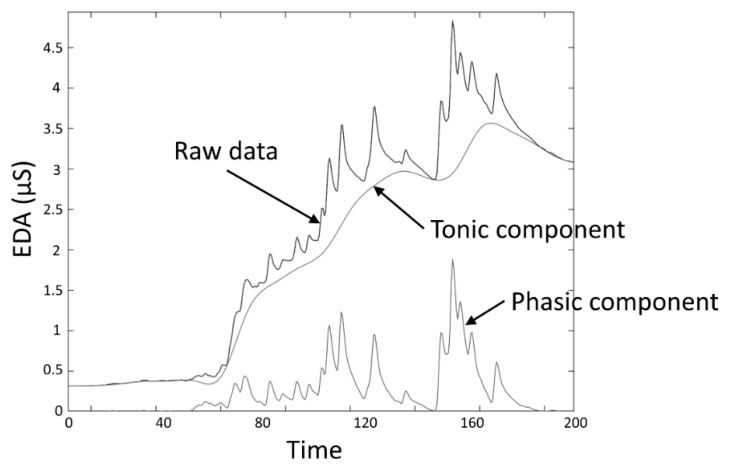
EDA data decomposition into tonic and phasic components.

**Figure 4 sensors-20-00479-f004:**
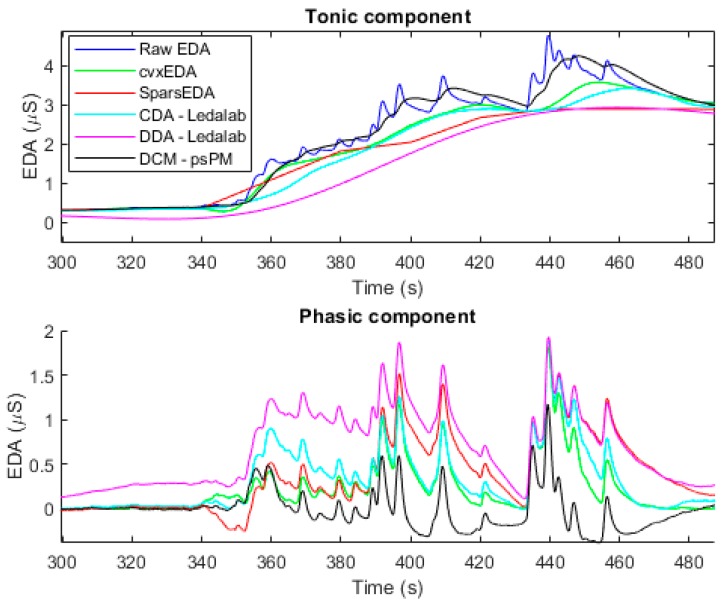
Tonic/phasic decomposition of a sample EDA signal using some available tools.

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
