# Peer review of "Innovations in Electrodermal Activity Data Collection and Signal Processing: A Systematic Review"

_sensors, 2020, doi:10.3390/s20020479_

Round 1

Reviewer 1 Report

This paper presents a systematic review of past and recent innovations in EDA data collection and signal processing that is quite detailed and thorough. Most of the significant publications and recording techniques in EDA are covered in this review. The authors could improve this work by categorically subdividing the topics covered in the Results section to clearly differentiate between different EDA recording and processing methods.

Some useful edits are suggested below:

Line 31: EDA does have a strong correlation with sweat production but for most contexts, it is related to psychophysiological or emotional sweating. Could the authors clarify the distinction between thermoregulatory sweat gland activity (mentioned in Line 38) and emotional sweating activated by the sympathetic nervous system earlier in the introduction? SCR is referenced in line 58 but defined in line 60-62. Please define before reference. Please include axis units on all plots. Figure 2 could be improved to show the relative amplitude and time-scale that an SCR is observed. Could the authors include units on the X and Y axis and improve their description of the signal characteristics to include the signal bandwidth, range of measurement, and smallest measurable SCR amplitude. Consider combining Fig. 2 with Fig. 3 characterize the EDA signal. Line 106: Please clarify the difference between endosomatic and exosomatic recording techniques using two distinct paragraphs or sections. Endosomatic EDA recording is not as common as exosomatic recording throughout literature. Could the authors address these recording techniques separately and comment on why this is the case? In general, the results section 3.1 needs to be subdivided to clarify the different recording methods (ex. endosomatic vs exosomatic, AC vs DC, etc.) Line 351: Move URL reference in References section. Please expand on the challenges associated with exosomatic EDA measurement using an AC and DC system. Why would a sensor designer choose one method over the other? What are the limitations or benefits for each method? Please expand on the inter-subject variability of time-domain EDA measurement in line 380. This is an important factor to consider when purchasing or designing an EDA sensor. In the Discussion section, please elaborate on what recording technologies and/or techniques that are most relevant to future applications of EDA.
The Conclusion could be strengthened given the material covered.

Author Response

We want to thank the reviewer for the comprehensive revision of our manuscript.  We have included a list of updates we have made to the manuscript, considering your comments and suggestions.  Responses and changes made to the manuscript are highlighted.

Comments:

This paper presents a systematic review of past and recent innovations in EDA data collection and signal processing that is quite detailed and thorough. Most of the significant publications and recording techniques in EDA are covered in this review. The authors could improve this work by categorically subdividing the topics covered in the Results section to clearly differentiate between different EDA recording and processing methods.

Response:

Thank you for your excellent suggestion.  We have added a more detailed categorical subdivision of the topics covered in the Results section.  We hope this clarifies the different topics covered by the reviewed literature.

Comment:

Some useful edits are suggested below:

Line 31: EDA does have a strong correlation with sweat production but for most contexts, it is related to psychophysiological or emotional sweating. Could the authors clarify the distinction between thermoregulatory sweat gland activity (mentioned in Line 38) and emotional sweating activated by the sympathetic nervous system earlier in the introduction?

Response:

Thank you for your comment.  We have modified the text earlier in the introduction to clarify the distinction between thermoregulatory sweat gland activity and emotional sweating. 

Comment:

SCR is referenced in line 58 but defined in line 60-62. Please define before reference.

Response:

Thanks. We have defined the term SCR the first time it is used.

Comment:

Please include axis units on all plots.

Response:

Thank you for your suggestion.  To improve the clarity of the figures, we have included axis units on all plots.

Comment:

Figure 2 could be improved to show the relative amplitude and time-scale that an SCR is observed.

Response:

Thank you.  Figure 2 now includes the relative amplitude and time scale with respect to stimulus time.

Comment:

Could the authors include units on the X and Y axis and improve their description of the signal characteristics to include the signal bandwidth, range of measurement, and smallest measurable SCR amplitude.

Response:

Thank you for this important suggestion.  We have added X and Y axes to all the figures.  Furthermore, we have added a paragraph with a description of the characteristics of the signal with the normal ranges, as follows:

“Despite the source that caused the SCRs (specific to a stimulus or spontaneous), they are characterized by a rise from initial level to a peak, followed by a decline [18].  When caused by a stimulus, the onset of the SCR is typically between 1 and 5 seconds after the delivery of the stimulus [3]. The amplitude of the SCRs (conductance at the peak relative to the conductance at the onset) can reach several µS. A minimum of 0.05 or 0.04 µS is typically set as a threshold to define a significant SCR, to avoid incorrect measurements caused by movement artifacts, the noise level of the equipment, and experimental conditions [3]. The time from the onset of the SCR to the peak, termed rise time (Figure 2), normally varies between 0.5 and 5 s [19]. The spectral content of EDA is mostly confined to 0.045–0.15 Hz [20]. Exercise increases the spectral content of EDA, exhibiting spectral content at about 0.37 Hz when subjects perform vigorous-intensity exercise [21]..”

Comment:

Consider combining Fig. 2 with Fig. 3 characterize the EDA signal.

Response:

We appreciate your suggestion. However, Figure 3 is intended to show the EDA in a broad sense, showing the tonic and phasic components of a multi-minute EDA recording, and Figure 2 is a detailed description of the characteristics and measures of a specific SCR, which comprises about 10 seconds.  We believe that combining those figures could muddle clarity.

Comment:

Line 106: Please clarify the difference between endosomatic and exosomatic recording techniques using two distinct paragraphs or sections. Endosomatic EDA recording is not as common as exosomatic recording throughout literature. Could the authors address these recording techniques separately and comment on why this is the case? In general, the results section 3.1 needs to be subdivided to clarify the different recording methods (ex. endosomatic vs exosomatic, AC vs DC, etc.)

Response:

Thank you for this comment.  We have created a subsection called “3.1.1. Endosomatic vs Exosomatic recordings,” and addressed these recordings in two separate paragraphs, for more clarity.  Furthermore, we have commented on why endosomatic recordings are less common than exosomatic recordings.  We have also created subsections “3.1.2.         DC vs. AC sources in exosomatic recordings,” “3.1.3.  Basic considerations on electrodes for EDA,” and “3.1.4. Advances in technologies for EDA data collection.”  The latter specifically includes the most recent publications in the field.

Comment:

Line 351: Move URL reference in References section.

Response:

Thank you for the suggestion.  We have moved this URL to the references section.

Comment:

Please expand on the challenges associated with exosomatic EDA measurement using an AC and DC system. Why would a sensor designer choose one method over the other? What are the limitations or benefits for each method?

Response:

Thank you for your comment.  We have extended the subsection “DC vs. AC sources in exosomatic recordings” to include the benefits and disadvantages of both approaches in exosomatic recordings.

Comment:

Please expand on the inter-subject variability of time-domain EDA measurement in line 380. This is an important factor to consider when purchasing or designing an EDA sensor.

Response:

Thank you for your suggestion.  We have provided a more detailed explanation of the inter-subject variability, as follows:

“Specifically, the SCL and NS.SCRs exhibited higher coefficient of variation (i.e., the standard deviation of measurements divided by the mean) and lower degree of consistency (assessed using intra-class correlation coefficient) between subjects undergoing cognitive, postural, and physical stress, when compared to spectral indices.”

Comment:

In the Discussion section, please elaborate on what recording technologies and/or techniques that are most relevant to future applications of EDA.

Response:

Thank you for this suggestion.  We have elaborated on the recording technologies and techniques more relevant for future developments of applications based on EDA.

Comment:

The Conclusion could be strengthened given the material covered.

Response:

We appreciate your comment.  We have strengthened the conclusion based on the material covered.

“The best electrode type for EDA data collection highly depends on the application.  Although Ag/AgCl hydrogel electrodes provide the most reliable signals, they are impractical for wearable applications that require long-duration reliability. More research is required to validate the feasibility of stainless steel and carbon electrodes to provide comparable signal quality.  Developing reusable electrodes reliable for long-term applications can boost the deployment of EDA applications.  As for the electronic devices, AC-voltage devices seems to be the most appropriate approach, overall. Despite their higher circuit complexity, they avoid the issues caused by electrode polarization. Current techniques for automatic analysis of EDA signals still require improving their subject independency and their ability to function without setting any external parameters, to make the technique feasible for ambulatory settings.  Furthermore, alternative techniques like spectral analysis have emerged as potential tools for the analysis of EDA.  Some tools for detecting and removing corrupted segments from EDA signals are available, but their sensitivity needs to be further evaluated. The consistency of the measures derived from EDA is still a matter of concern, specifically in the presence of motion artifacts..”

Reviewer 2 Report

It is nice to have on overview paper of the last 10 years on the EDA signal processing and instrumentation. The manuscript here gives a good overview on done work on the chosen topics.

However, I got the impression that the paper here is like a listing of references, in which 1 or 2 sentences of each paper are picked out and rewritten within this paper. In my personal point of view I would have liked this paper to bring the work that is done more together into context.

For example, just writing that "the main advantage of AC-source devices is avoiding electrode polarization [3]" is too little. As stated in [3] or other references, the AC method has the advantage that skin potential, skin conductance and susceptance can be simultaneously measured at the same skin site. All these three quantities together might give more information. There are actually groups doing research on this. The authors here mentioned only that it is possible to measure skin potential and conductance at the same time a little bit later in the text (line 135-136).

-In terms of electrodes I am missing statements about conductive gel vs. dry electrodes.

-The authors should double check that the written sentences reflect the content of each reference correctly. Some statements are not correct or misleading:

For example. It is written that in lines 137 to 139 that "Another study looked at the feasibility of high-frequency (up to 70 kHz) alternating current for collecting EDA [21]. The high-frequency measurements overestimated the EDA by about 20%, but could still be used to allow higher sampling rates of EDA." It is true that measurements with higher excitation frequencies allow for higher sampling rates of EDA. However, the "up to 70kHz" is wrong. The authors of [21] did measurements up to 70kHz, but only the 500Hz to 10 kHz interval was used to extrapolate to lower frequencies by the model.

Another example is reference [22]. The sentence (lines 139 - 140) "A more recent study evaluated the error due to electrode polarization produced by DC-source devices, even with nonpolarizing electrodes [22]" This sentence does not reflect the objective of the study. The study in [22] was meant to directly compare the DC vs. the AC method of recording EDA. The DC method is more common since it is easier to implement. The authors in [3] mentioned that the AC method is superior to the DC method due to several reasons but stated that also more research is needed.

It seems like that the authors did a lot of work on EDA signal processing themselves and have good expertise in this part. I would like the authors to read some of the references again and correct statements in parts of the paper that they are not so familiar with.

Author Response

We want to thank the reviewer for the comprehensive revision of our manuscript.  We have included a list of updates we have made to the manuscript, considering your comments and suggestions.  Responses and changes made to the manuscript are highlighted.

Comment:

It is nice to have on overview paper of the last 10 years on the EDA signal processing and instrumentation. The manuscript here gives a good overview on done work on the chosen topics.

However, I got the impression that the paper here is like a listing of references, in which 1 or 2 sentences of each paper are picked out and rewritten within this paper. In my personal point of view I would have liked this paper to bring the work that is done more together into context.

Response:

Thank you for your comment.  We have provided more context to the description of the revised works throughout the results section.

Comment:

For example, just writing that "the main advantage of AC-source devices is avoiding electrode polarization [3]" is too little. As stated in [3] or other references, the AC method has the advantage that skin potential, skin conductance and susceptance can be simultaneously measured at the same skin site. All these three quantities together might give more information. There are actually groups doing research on this. The authors here mentioned only that it is possible to measure skin potential and conductance at the same time a little bit later in the text (line 135-136).

Response:

Thank you for this comment.  We have expanded on the advantages and disadvantages of the AC and DC methods in the results and discussion sections, including the capabilities to collect SP, SC, and SS at the same skin site.

Comment:

In terms of electrodes I am missing statements about conductive gel vs. dry electrodes.

Response:

Thank you for this relevant comment.  We have extended the text about advantages and disadvantages of hydrogel and dry electrodes in the results section, as follows:

“Due to the nature of EDA and the location where the signal is acquired, recording EDA signals may require special electrodes, electrode gels, and recording devices that differ from the ones used for other psychophysiological measures.  Electrodes are an important factor in the quality of the EDA measures.  The use of the same material for the two electrodes is very important for EDA recordings, mainly in DC-source devices, because a difference in metals will introduce a potential difference. The use of silver-silver chloride (Ag/AgCl) electrodes is the standard for EDA recording.  Ag/AgCl electrodes help minimize both polarization of the electrode and the bias potential between the electrodes.  The hydrogel improves the signal quality by lowering the impedance that exists at the electrode–skin interface.  However, it reduces the shelf life of hydrogel electrodes as the hydrogel layer that exists in the skin–electrode interface degrades with time due to dehydration.  The high impedance produced by dehydration leads to an impairment in the quality of the signal and to an increased signal corruption caused of motion artifacts and noise.

              Dry electrodes can potentially avoid the signal degradation issues and inventory shelf life management. Metallic electrodes are widely used, but carbon electrodes have gained some popularity for EDA recordings [3].  In general, the main advantage of dry electrodes over hydrogel electrodes is that they do not degrade over time (lacking hydrogel), and thus can potentially collect better long-term recordings. As dry electrodes do not have a shelf life limitation, they also save the expense of managing inventory and scrapping obsolete inventory. 

A big concern in EDA measurement is the polarization of the electrodes.  In exosomatic DC-source devices, the electrodes are polarized by electrolysis because they carry a DC current and become anode and cathode in an electrical system.  Reusable EDA electrodes, often commercially available, are much more expensive than disposable AgCl electrodes with very thin layers of silver chloride. In reusable electrodes, it has been observed that even with low DC current, after prolonged use the charge can become sufficiently high to remove part of the AgCl layer at the cathode, and increase the layer at the anode, producing large bias voltage levels [22]. “

Comment:

The authors should double check that the written sentences reflect the content of each reference correctly. Some statements are not correct or misleading:

For example. It is written that in lines 137 to 139 that "Another study looked at the feasibility of high-frequency (up to 70 kHz) alternating current for collecting EDA [21]. The high-frequency measurements overestimated the EDA by about 20%, but could still be used to allow higher sampling rates of EDA." It is true that measurements with higher excitation frequencies allow for higher sampling rates of EDA. However, the "up to 70kHz" is wrong. The authors of [21] did measurements up to 70kHz, but only the 500Hz to 10 kHz interval was used to extrapolate to lower frequencies by the model.

Response:

We apologize for the misleading statement.  We have checked the results section throughout. We have corrected the specific paragraph as follows:

“Another study looked at the feasibility of high-frequency alternating current for collecting EDA [25]. They collected skin admittance measurements from 1 Hz to 70 kHz, and used the interval 500 Hz to 10 kHz to fit a Cole model to the measured skin admittance. The method overestimated the skin conductance by about 20%, suggesting that the proposed skin admittance method is not suitable for estimation of low frequency skin conductance level, but the method can still be used to collect the variations of EDA at higher sampling rates.  ”

Comment:

Another example is reference [22]. The sentence (lines 139 - 140) "A more recent study evaluated the error due to electrode polarization produced by DC-source devices, even with nonpolarizing electrodes [22]" This sentence does not reflect the objective of the study. The study in [22] was meant to directly compare the DC vs. the AC method of recording EDA. The DC method is more common since it is easier to implement. The authors in [3] mentioned that the AC method is superior to the DC method due to several reasons but stated that also more research is needed.

Response:

We have also revised this paragraph so its content is in agreement with what was reported in the referenced manuscript, as follows:

“A more recent study aimed to directly compare AC and DC measurements of EDA, following the suggestion from [3], in order to validate the AC method by comparing it to a standard DC method [26].  They found a voltage of 0.2 V to be sufficient for DC recordings. Their main observation was the excellent agreement they found between a 20 Hz AC method and a standard DC method.  These studies provide further support of the validity and superiority of the AC recording methodology.”

Comment:

It seems like that the authors did a lot of work on EDA signal processing themselves and have good expertise in this part. I would like the authors to read some of the references again and correct statements in parts of the paper that they are not so familiar with.

Response:

Thank you for your very important comment.  We have read in more detail the references we were not as familiar with, and corrected the statements when necessary.

Reviewer 3 Report

An interesting manuscript that provides a reasonable review of EDA signal processing and analysis. The manuscript can be more informative if authors at a section that includes the application of EDA in other domains such as the construction industry. They are several quality papers published on this topic to analyze EDA signals in a noisy environment such as construction sites.

Author Response

We want to thank the reviewer for the comprehensive revision of our manuscript.  We have included a list of updates we have made to the manuscript, considering your comments and suggestions.  Responses and changes made to the manuscript are highlighted.

Comment:

An interesting manuscript that provides a reasonable review of EDA signal processing and analysis. The manuscript can be more informative if authors at a section that includes the application of EDA in other domains such as the construction industry. They are several quality papers published on this topic to analyze EDA signals in a noisy environment such as construction sites.

Response:

Thank you for your comment.  We focused this review on manuscripts that proposed novel approaches for collecting and processing EDA, and limited applications of EDA to physiological systems.  Although the blossoming of applications of EDA motivated us to write the present review, we aimed to provide some guidance to researchers pursuing any application (for instance, stress assessment in construction sites) about the best technologies so far for EDA data collection, and what techniques are available for EDA data processing.  As for the applications of EDA during the last 10 years, we found they are multiple and diverse (attention, stress, PTSD, depression, emotion recognition, impulsivity, sleep monitoring, kleptomania, psychopathy, gambling, aggressive behavior, fear, neuropathies, exercise, COPD, muscle fatigue, epilepsy, advertising perception, etc.), and reviewing them could be a demanding endeavor, and will result in an extensive review by itself. Although such review is scientifically relevant for the field, it falls out of the scope of the present review.

Reviewer 4 Report

The authors summarize and review the recent developments  and provide an updated and synthesized framework for old and new researchers interested in  incorporating EDA into their research.

The conclusions are poor. To improve the paper the authors should extend the conclusions.

Author Response

We want to thank the reviewer for the comprehensive revision of our manuscript.  We have included a list of updates we have made to the manuscript, considering your comments and suggestions.  Responses and changes made to the manuscript are highlighted.

Comment:

The authors summarize and review the recent developments  and provide an updated and synthesized framework for old and new researchers interested in  incorporating EDA into their research.

The conclusions are poor. To improve the paper the authors should extend the conclusions.

Response:

Thank you for your comment.  We have extended the conclusion of the paper to better reflect the findings of the extensive review performed.  The new content in the conclusion section is highlighted.

“The best electrode type for EDA data collection highly depends on the application.  Although Ag/AgCl hydrogel electrodes provide the most reliable signals, they are impractical for wearable applications that require long-duration reliability. More research is required to validate the feasibility of stainless steel and carbon electrodes to provide comparable signal quality.  Developing reusable electrodes reliable for long-term applications can boost the deployment of EDA applications.  As for the electronic devices, AC-voltage devices seems to be the most appropriate approach, overall. Despite their higher circuit complexity, they avoid the issues caused by electrode polarization. Current techniques for automatic analysis of EDA signals still require improving their subject independency and their ability to function without setting any external parameters, to make the technique feasible for ambulatory settings.  Furthermore, alternative techniques like spectral analysis have emerged as potential tools for the analysis of EDA.  Some tools for detecting and removing corrupted segments from EDA signals are available, but their sensitivity needs to be further evaluated. The consistency of the measures derived from EDA is still a matter of concern, specifically in the presence of motion artifacts.”

Round 2

Reviewer 1 Report

Thank you for your improvements to this manuscript. The Authors have sufficiently addressed all of the major revisions recommended since their first submission and this latest edition provides a solid contribution to this field of research.